# Extensive photochemical restructuring of molecule-metal surfaces under room light

Chenyang Guo[1], Philip Benzie[1,2], Shu Hu [1], Bart de Nijs [1], Ermanno Miele [1], Eoin Elliott[1], Rakesh Arul [1], Helen Benjamin[2], Grzegorz Dziechciarczyk[2], Reshma R. Rao [3], Mary P. Ryan[3] & Jeremy J. Baumberg [1]✉

The molecule-metal interface is of paramount importance for many devices and processes, and directly involved in photocatalysis, molecular electronics, nanophotonics, and molecular (bio-)sensing. Here the photostability of this interface is shown to be sensitive even to room light levels for specific molecules and metals. Optical spectroscopy is used to track photoinduced migration of gold atoms when functionalised with different thiolated molecules that form uniform monolayers on Au. Nucleation and growth of characteristic surface metal nanostructures is observed from the light-driven adatoms. By watching the spectral shifts of optical modes from nanoparticles used to precoat these surfaces, we identify processes involved in the photo-migration mechanism and the chemical groups that facilitate it. This photosensitivity of the molecule-metal interface highlights the significance of optically induced surface reconstruction. In some catalytic contexts this can enhance activity, especially utilising atomically dispersed gold. Conversely, in electronic device applications such reconstructions introduce problematic aging effects.

The stability of the metal–molecule interface is at the heart of many key processes which depend on surface binding and/or electron transfer. It is well known that pristine catalyst surfaces reconstruct under operational conditions[1–5], but this is challenging to identify in the first key step of photocatalysis even though adatom and edge sites typically support much higher chemical reactivity. By contrast, the electrical contact between conducting molecules and metal electrodes is known to be crucial for optimal device performance[6–9], but this has been hard to control and characterize under operational conditions. Adatoms can pick out specific molecules to focus current through and induce diode behaviour[10]. For nanophotonic devices, the stability of the metal–molecule surface is assumed under ambient light, and while it is known that laser illumination can cause degradation, the mechanisms for this remain unclear.

In all such cases, surface atom stability is crucial. In many molecular devices, gold is used as a preferred electrode due to both its high electrical conductivity and chemical stability. While studies using scanning tunnelling microscopy[11] show that gold atoms on a gold surface are not completely stable, adatoms can only be induced by powerful forces (>1 nN)[12] such as strong electric fields, and do not appear when the temperature is low enough[13] or in vacuum[14]. On the other hand, molecules binding to a Au surface are known to induce localized reconstruction. For instance, the strong Au-thiol bond has been found to create non-metallic Au(I) 'staples' which bridge neighbouring sulphur atoms[15], and similar effects are found for any strong metal–molecule bond. Recently it was found that optical forces producing metal adatoms can be thousands of times larger than expected when under the extreme electromagnetic fields inside plasmonic cavities[12]. However, it remains unclear how this relates to untextured standard metal surfaces used in devices.

Here, we examine the photostability of molecular-coated Au surfaces. Many molecular electronics and optomolecular devices use thiol contacting which is known to produce extremely stable and uniform self-assembled monolayers (SAMs)[16]. To ensure relevance for

[1]Nanophotonics Centre, Department of Physics, Cavendish Laboratory, University of Cambridge, Cambridge CB3 0HE England, UK. [2]Cambridge Display Technology Ltd, Cardinal Way, Godmanchester PE29 2XG, UK. [3]Department of Materials, Imperial College London, London SW7 2AZ, UK. ✉e-mail: jjb12@cam.ac.uk

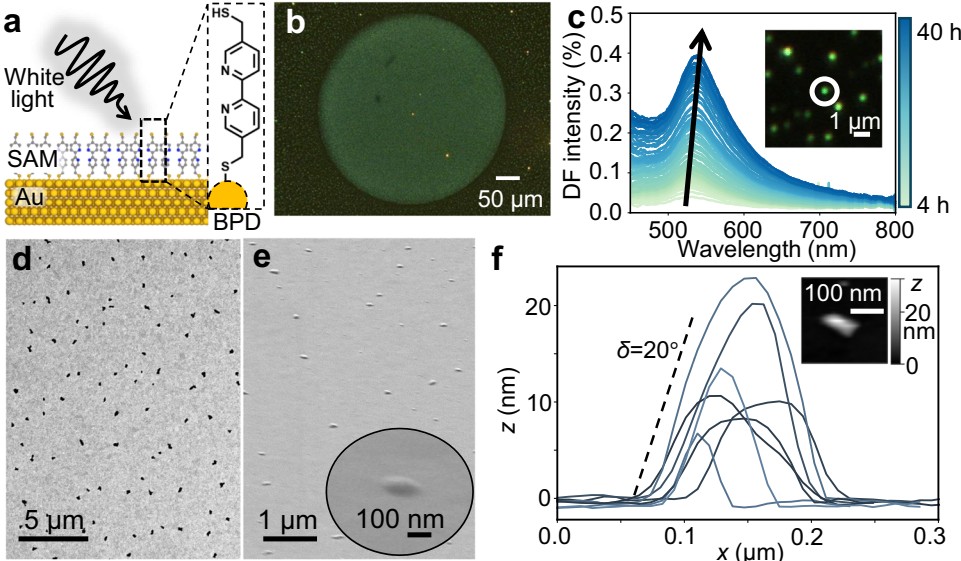

**Fig. 1 | Nanocaps produced by white light irradiation of BPD molecular layer on gold. a** Schematic of 2,2′-BPD layer on gold illuminated by white light. **b** Nanocaps formed within illuminated area after 10 h of white light exposure (illuminated area seen as central brighter region). **c** Single nanocap dark-field scattering spectra which redshifts with time during irradiation. **d, e** SEM images of photocreated nanocaps on Au from (**d**) top and (**e**) 45° angle view. inset: SEM image of a single nanocap. **f** AFM profiles of several nanocaps, inset: AFM topographic image, greyscale shows height.

optomolecular devices, all measurements are performed under ambient conditions and at room temperature. In these conditions, thiol SAMs exclude contamination and provide a well-defined geometry (see Methods for optimization and characterization). We find a particularly light-sensitive molecule, 5,5-*bis*(mercaptomethyl)−2,2-bipyridine (2,2′-BPD, inset Fig. 1a), which forms stable SAMs on Au at room temperature in the absence of light but on illumination with even very weak light (0.1 W cm$^{-2}$) shows strong photosensitivity. This results in the formation of defects across the BPD-coated Au surface that are clearly observable in dark-field (DF) optical microscopy. We attribute the origin of these defects to the formation of Au 'nano-caps' due to the lateral diffusion of gold atoms plucked from the underlying Au substrate through the SAM. We find that other molecules also photo-destabilize the flat Au at higher intensities, and present measurements on an alternative nanostructure that allows improved quantitation of the photo-diffusion by spectral shifts, enabling direct comparison between different molecular SAMs.

## Results
### Nanocap formation and light-induced dynamics
Thiol SAMs are prepared on template-stripped Au which is locally atomically flat (Methods). We first discuss 2,2′-BPD which has a core of 2,2-bipyridine, a ligand that strongly binds metal ions[17], and is attached by methyl to thiol groups. This molecule is known to crosslink under electron beam excitation[18]. Initially, samples are placed under a microscope and irradiated by incandescent broadband white light through a 100 × 0.9 NA objective lens at intensities of 10 W cm$^{-2}$ (Fig. 1a). Green spots from each nanocap are seen to gradually grow in number and intensity within the ≈400 µm diameter illuminated area (Fig. 1b, imaged by a ×20 lens). Dark-field scattering spectra on each individual nanocap under 0.1 W cm$^{-2}$ reveals that they all show a peak initially at $\lambda$=520 nm which redshifts linearly with time at 2.4 nm h$^{-1}$ W$^{-1}$ cm$^2$ (Fig. 1c). Without light, no change is seen (even over weeks) while without the thiol molecular layer, no change in the template-stripped Au is observed. Control experiments eliminate effects from the underlying substrate and thermal effects. Nanocaps form in the same way on gold-coated silicon wafers (Supplementary Fig. 12). On the other hand, heating the samples to 100 °C in the absence of light does

not result in nanocap formation (Supplementary Fig. 13), affirming that it is a light-induced process.

### Surface morphology from scanning microscopies
To understand the surface morphology of the nanocaps, they are probed by scanning electron- and atomic force microscopies (SEM, AFM). Top-view SEMs reveal the nanocaps exhibit an irregular shape, ranging from 50-400 nm in diameter. Regions that appear darker in SEM images indicate areas with lower emissivity of secondary electrons, normally from regions of reduced electrical connectivity[19–25]. These observations thus suggest that a continuous molecular layer remains beneath the nanocaps (Fig. 1d, e). High-angle backscattered electron images confirm that their composition is predominantly gold (see Supplementary Fig. 3). AFM confirms that the nanocaps all have similar aspect ratio with a slope angle of $\delta$=20° (Fig. 1f) but grow in volume over time. This constant Au-SAM wetting angle (which is very different to typical Au islanding) implies it is controlled by the surface energy of the Au-SAM interface compared to Au-air ($U_{\text{Au/air}}$≈0.88 eV nm$^{-2}$)[26,27]. Since Au-S binding energies are 1.95 eV per molecule, the SAM area of 0.26 nm$^2$ per molecule gives an estimated surface energy $U_{\text{Au/SAM}}$≈0.8 eV nm$^{-2}$, which reasonably matches $U_{\text{Au/air}} \cos \delta$≈0.9 eV nm$^{-2}$. We note that once nanocaps form, they are pinned in place and do not move.

### Nanocap spatial distribution and dynamics
To better understand nanocap formation, we use thresholding-based image analysis[28,29] to locate the position of nanocaps in each image through a time series of dark-field images over $t$=0 to 63 h at 0.1 W cm$^{-2}$. Once the position of each nanocap is determined, detailed information about its optical intensity, size, and Voronoi area are extracted. The Voronoi construction segments an area around each nanocap using the perpendicular bisectors of lines to all neighbours (Fig. 2d inset). The distribution of distances $r$ to their nearest neighbours (inset Fig. 2a) reasonably matches a Weibull distribution $p(r) = n2\pi r \exp(-\pi n r^2)$ for density $n$, which is expected for a random spatial arrangement[30]. Their characteristic separation $r_0$ found at each time $t$, reduces as more nanocaps form in-between, with $r_0 \propto t^{-1}$.

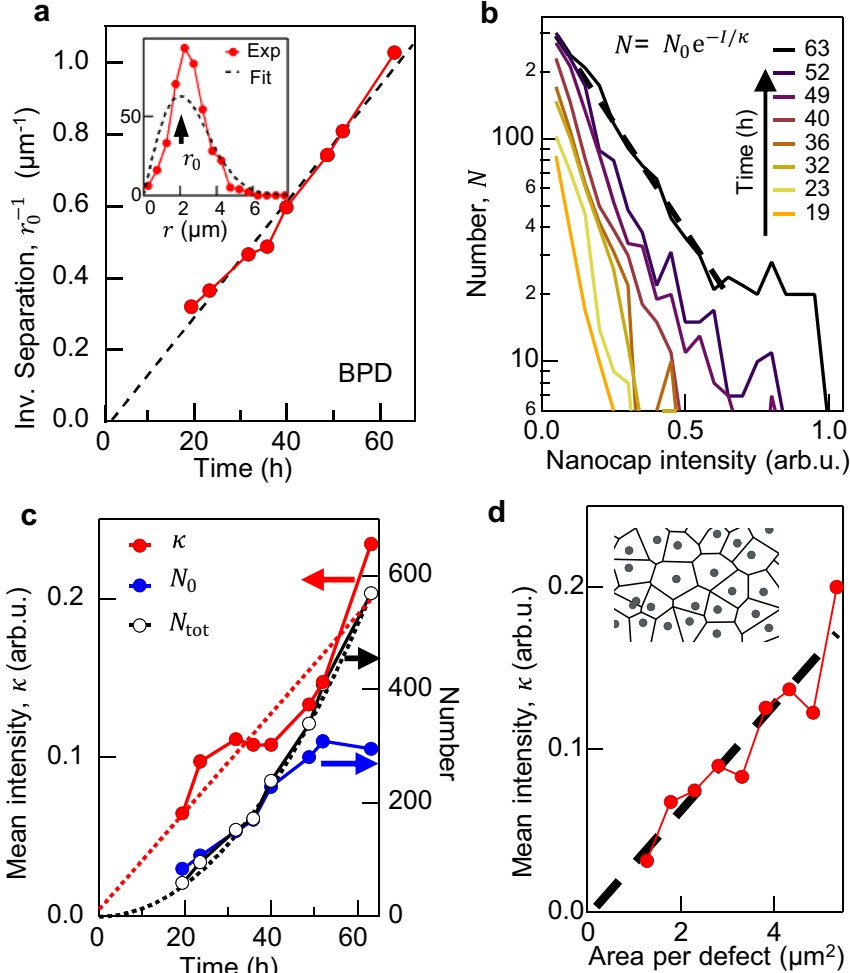

**Fig. 2 | Growth dynamics of nanocaps on BPD-Au. a** Inverse separation between nanocaps increases linearly with time. Inset: characteristic separation $r_0$, from fit to Weibull distribution of nearest separations at $t=32$ h. **b** Distribution of nanocap scattering intensities is exponential, with slope $\kappa$ which evolves in time. **c** Extracted $\kappa, N_0$ as well as total number of nanocaps (black) versus time. **d** Mean nanocap intensity versus Voronoi area surrounding each one, after $t=63$ h. Inset: Voronoi construction for extracting areas (see text).

The distribution of nanocap scattering intensities at each time (Fig. 2b) is found to follow an exponential distribution, $N = N_0 e^{-I/\kappa}$ (dashed line). Initially all nanocaps are small and so scatter weakly, with scattering typically $I \propto V^2$ for volume $V \ll \lambda^3$ (Rayleigh scattering). For $\kappa$ to increase with time, some nanocaps must grow much faster than others. We find that $\kappa$ increases linearly with time, while $N_0$ also increases near linearly (Fig. 2c). The total number of nanocaps thus increases roughly quadratically (black curve) with time since $N_{tot} = N_0\kappa$, in agreement with $N_{tot} \propto r_0^{-2} \propto t^2$. We also find a strong correlation between the intensity of each nanocap and the area $A$ surrounding it (Fig. 2d) which strongly supports a surface mobility model where photo-mobilized Au atoms are captured by nanocaps. On the other hand, no Ostwald ripening is observed as nanocaps are never found to shrink. At the same time, the quadratic rise in total number with time together with their random location suggests the seeding process depends on two-particle collisions between mobile Au atoms. The influence of the 'collection area' surrounding each nanocap (Fig. 2d) implies that once optically activated, the lateral atom diffusion is rapid over many microns.

**Photoactivity and nanocap formation model**
We thus suggest a model for the photoactivity (Fig. 3), which is based on light-induced forces at the molecule–metal surface which create adatoms and then lift them off the surface where they can move

laterally. For 2,2'-BPD, we hypothesize that the diffusive Au lies in N-coordinated sites as seen for observations with $Ni^{2+}$ [18,31], however Au-N coordination remains so far lacking direct experimental evidence. Pre-soaking the samples in $NiCl_2$ solution which fills these coordination sites with $Ni^{2+}$, is found here to turn off all Au photoactivity and produces no nanocaps. The diffusing adatoms which collide have a chance of emerging on the top surface of the SAM (possibly most easily at defects in the SAM polycrystalline molecular order). Note that no correlation is observed between nanocap locations and the crystal plane boundaries of the underlying Au substrate (seen as grey shades in SEM, Fig. 1d), suggesting Au atomic order underneath is less relevant. Once a nucleus is formed on the top of the SAM, the probability of mobile Au atoms surfacing there is high (as seen by their collection from a wider area, Fig. 3b), and thus the nanocaps grow, with morphology determined by the surface energy. This also explains why nanocaps do not seed close to each other (suppressed $f(r)$ for $r \ll r_0$) since adatom diffusion to an existing nanocap is more likely.

The initial stage of atom photo-extraction from the Au substrate appears to be a crucial step. Recently a mechanism was discovered that amplifies optical forces at metal surfaces a thousand-fold, which then becomes capable of creating Au adatoms [12]. Their creation rate $R \propto \exp\{-U(I)/k_B T\}$ depends on the energy barrier $U(I) = U_0(1 + I/I_t)^{-1}$, which is reduced with increasing laser intensity. The threshold intensity $I_t$ is inversely proportional to the local optical polarisability of the

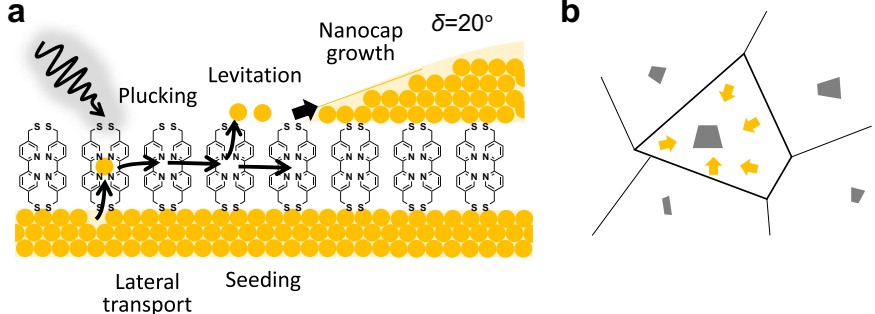

**Fig. 3 | Proposed mechanisms for light-driven Au migration. a** (I) Initial photo-induced plucking of Au atom from substrate which is lifted into a diffusive layer within the SAM. (II) Collisions between Au atoms increase probability for their levitation on top of the SAM where mobility is low. (III) Further Au atoms colliding with this nucleus grow the nanocap, with morphology set by surface wetting. **b** Collection area feeding growth of each nanocap, extracted from Voronoi image analysis to define formation area. Yellow arrows indicate movement of Au atoms, grey areas represent nanocap formation.

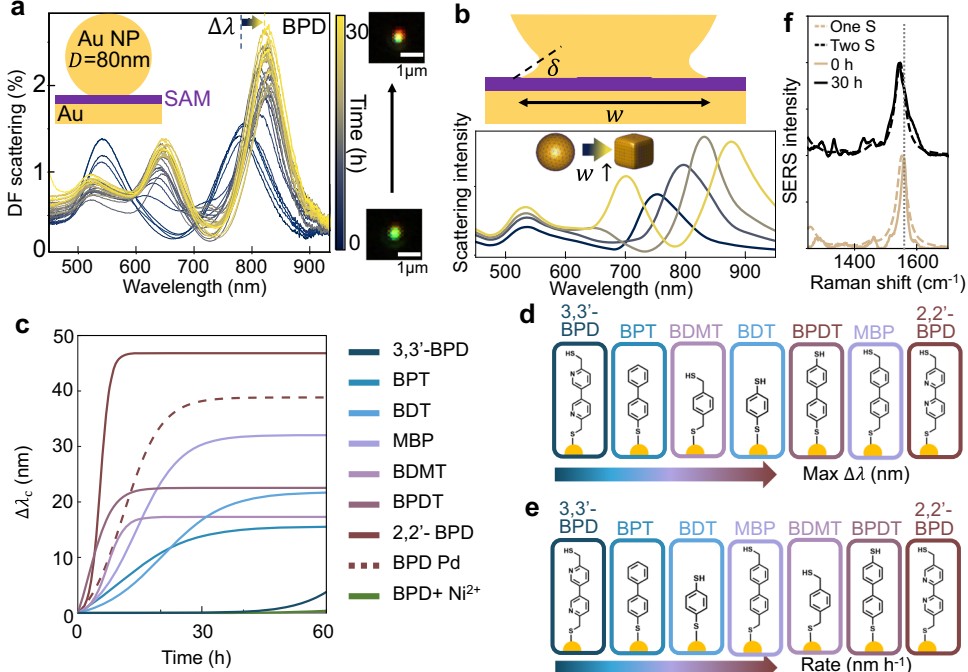

**Fig. 4 | Molecular photo-induced spectral shifts in NPoMs. a** DF scattering spectra of 2,2′-BPD in NPoM gap evolving over t=0-30 h under 0.1 W cm⁻², colour gradient from dark blue (start) to light yellow (end) gives time progression. **b** Simulation of NPoM scattering using BEM model, for nanoparticle facet $w$ increasing from 12 to 50 nm. **c** Spectral shift of coupled plasmon peak $\Delta\lambda_c$ versus time for seven different molecules: BPD, BPT, MBP, BDMT, BDT, BPDT in NPoMs, as well as for BPD with Pd-coated Au, and with Ni²⁺ pre-treatment. **d**, **e** Observed (**d**) maximum $\Delta\lambda_c$ and (**e**) rate $\dot\lambda_c$, in increasing rank. **f** Experimental SERS spectra (solid) and DFT Raman simulations (dashed) of BPD SAM in NPoM before (brown) and after (black) 30 h illumination at 0.1 W cm⁻². DFT uses one (brown dashed) or two (black dashed) thiol-Au terminations of BPD molecule. The grey dashed line marks the shifting bipyridine peak.

atom closest to the Au, which produces the enhanced dipole-dipole forces that drive this optical plucking. The evidence shown above suggests that the same mechanism is involved here, with BPD exhibiting the lowest $I_t$ yet known. Comparing the nanocap formation rate for white light excitation of 1–10 W cm⁻² (Supplementary Figs. 1, 2) shows it follows the same power dependence as $R$, however with $I_t$ more than thousand-fold smaller than for bi-phenyl-thiol (biphenyl-4-thiol, BPT). We thus examine a range of thiolated SAMs, but find that only 4,4-bis(mercaptomethyl)-biphenyl (MBP) also shows this photo-activity on flat Au and much more weakly.

## Quantitative investigation using NPoM structures

To thus better comprehend the different processes, we now exploit an alternative nanostructure to measure photo-dynamics for a wider range of molecules. Essentially this seeds nanocap formation by from the start placing Au nanoparticles (diameter 80 nm) on top of the SAM-coated Au, thus creating a nanoparticle-on-mirror (NPoM) plasmonic cavity[32]. At the metal–molecule interface, the light is now confined in the nm-sized gap between the facets, increasing inelastic light scattering by more than eight orders of magnitude[33]. The resulting DF scattering and surface-enhanced Raman spectra (SERS) help reveal changes in both metal and molecule nanostructure.

We first probe the DF scattering spectra which show strong peaks from the dominant coupled plasmon mode ($\lambda_c \approx 800$ nm) as well as shorter wavelength higher-order modes[33]. Automated sample translation and custom software allow repeated collection at each $t$ of DF spectra from the same ≈50 NPoMs, to periodically sample the distribution. From histograms of these spectra at each $t$, the modal representative spectrum is shown (Fig. 4a), evolving from two initial peaks (blue) to three peaks (yellow). The dominant coupled mode

redshifts by $\Delta\lambda_c$ and grows in intensity, captured directly in the visible DF images evolving from green to red halo/yellow spot.

Such red shifts can arise from morphological changes of the nanoparticle and/or the gap contents. Previous work has shown typical red shifts arise from expansion of the lower facet size $w$[34,35]. We thus simulate the evolution of the facet using a BEM model[36–38], under an initial assumption that the NP shape changes from sphere to cube. The DF scattering spectra are mostly sensitive to the gap region, so changes around the rest of the nanoparticle have little effect. These simulations show that as the NP facet becomes wider, the spectra indeed red-shift as detected in the experiment (Fig. 4b). An alternative explanation for the red-shift is a decrease in gap thickness or increase in refractive index of the SAM. While we discount these, we cannot discount these, the observed $\Delta\lambda_c \approx 48$ nm would require an increase in SAM refractive index from 1.50 to 1.75 from the intercalation of Au ions, or a decrease in gap thickness by 25%, both of which are less compelling explanations. Comparing the DF spectra of BPT and BPD (Supplementary Fig. 15) with simulations confirms gap sizes of $d = 1.3$ nm and 1.65 nm (respectively), well matching expectations for their near vertical orientation to the substrate.

### Comparative Study of Different SAMs

To compare the Au photo-migration for different molecular groups, the evolving DF spectrum of seven different SAMs are examined under 0.1 W cm$^{-2}$ white light (Fig. 4c–e, evolving spectra for each shown in Supplementary Fig. 5a–g, peak shifts for the three main DF modes shown in Supplementary Fig. 6). All samples are prepared in the dark in the same way as BPD, and SAM formation confirmed by DF and SERS. A redshift is seen for all molecules given enough time (Fig. 4c), due to facet growth (inset Fig. 4b, Supplementary Fig. 4) saturating at wetting angle $\delta$ set by the surface energies. This implies that without the AuNP present, an insuperable energy barrier exists in most SAMs, preventing initial seeding since they are found to never form nanocaps on their own. It also suggests that molecules such as biphenyl-4,4′-dithiol (BPDT) only move NP atoms to grow the facet rather than plucking them from the substrate. As expected, molecules with a thiol at the top have lower Au-SAM surface energies and redshift more (Fig. 4d). We also find longer molecules redshift more, suggesting flexibility of the SAM plays some role in surface energy, or that partial charge transfer is important. This is particularly notable for 2,2′-BPD versus MBP, showing the bidentate N-coordination is important.

On the other hand, some molecules (particularly BPD) redshift much more rapidly (Fig. 4e), suggesting that Au migration is faster, or that they have a reduced energy barrier for accumulating Au atoms. The comparison between 2,2′-BPD and 5,5′-bis(mercaptomethyl)-3,3′-bipyridine (3,3′-BPD, termed BPD′) reinforces the influence of molecular structure on the movement of gold atoms. If bipyridine-Au coordination is absent (with BPD′), motion is quenched, even compared to BPT. This suggests that charge transfer is crucial to enhance the light-induced barrier suppression discussed above, since the anionic Au(III)[BPD]$_2^-$ species can be tenfold more polarizable (compared for instance to the neutral Ni(II)[BPD]$_2$)[39]. Adatom plucking into this bipyridine site is thus accompanied by Au atom oxidation, with the diffusion of the resulting Au(III) accompanied by electron diffusion (either on the surface of the Au facets, or in the SAM).

### Theoretical Considerations and DFT Simulations

The above observations emphasize the challenge of theoretical calculations. Since $\Delta\lambda_c$ evidently depends on the flexibility of the SAM, DFT simulations must include many molecules and their interactions faithfully enough to capture surface energies at the top Au facet. Separately, since $\lambda_c$ depends on partial charge transfer, more complex DFT is required to deal with non-integral electronic redox. Time-dependent density-functional theory (TDDFT) is used to calculate the polarizability of 2,2′-BPD with 0, 1, 2 terminal thiol groups bound to Au

(Supplementary Figs. 9, 10), which we find increases with more Au coordination bonds (Supplementary Note 5). Adding these neutral Au atoms at the ends of the molecule (or between two BPD molecules) changes the resonant absorption of BPD, with a new charge-transfer absorption appearing in the visible (400–500 nm). This charge transfer resonance increases the polarizability of the molecule, enabling the action of light-induced forces. The difference between the bipyridine and biphenyl ring indeed shows that the bipyridine ring has larger polarizability at 700-800 nm (Supplementary Fig. 11), thus eliciting stronger optical-induced van der Waals forces. We note that full DFT calculations still find it hard to properly account for the effects of image charges in the close-by metal (within 0.5 nm), as well as intermolecular interactions and solvation in the constrained geometry of the nanogap.

### Control Measurements and Implications

The role of lateral migration is again confirmed for this NPoM geometry, by filling the bidentate sites using Ni$^{2+}$ pre-treatment, with no redshifts observed even after 50 h (Fig. 4c). This Ni$^{2+}$ control confirms that occupying the N-coordination sites halts Au migration. This also confirms that optical crosslinking is not involved since this is known to be rapid for Ni$^{2+}$ pre-treatment, thus confirming that refractive index or gap changes are not involved (Supplementary Fig. 5i). For another control, a monolayer of Pd atoms is electrochemically deposited on the Au mirror which doubles the energy needed to extract metal atoms[40]. In this case the maximum $\Delta\lambda_c$ is similar (similar wetting energy at the top of the molecule), but the growth rate is four-fold slower, limited by the rate of plucking out the metal atoms (Fig. 4c black dashed, Supplementary Fig. 5g).

To confirm the molecules are not degraded during this photomigration, SERS spectra recorded using a 633 nm laser at the start and end of scans (thus minimizing additional optical migration) show little reduction in signal. The strong bipyridine peak at 1556 cm$^{-1}$ is seen to shift to lower energies by $\approx$15 cm$^{-1}$ (Fig. 4f, Supplementary Fig. 8), which is predicted by DFT simulations when changing from BPD binding to Au at one end (brown), to thiol linkage with gold at both ends (black). This further supports our model for optical-induced Au migration atop the SAM since facet growth increases the number of molecules interacting with gold atoms, thus developing this vibrational shift. Additionally, the observed redshifts in the dark-field spectra of 2,2′-BPD NPs under various experimental conditions further corroborate these findings (Supplementary Fig. 7).

Our observations that even very low light intensities (comparable to room light) can optically reconstruct metal–molecule surfaces are relevant to many research fields. Since local polarizabilities drive the initial stage of the process, this understanding has predictive power. For instance, it is relevant for many metals beyond Au, Ni and Pd shown here, particularly where coordination bonds give partial charge transfer. It also explains why redox-active molecules or polymers can rapidly destabilize metal surfaces. Even where nanocaps do not form, plucking of surface atoms can trigger photocatalysis, electronic and spin transfer, or analyte binding.

### Discussion

In conclusion, this work shows that even weak light can extract gold atoms from a solid facet and force them to pass rapidly through a molecular layer to the upper surface. Lateral migration of these Au atoms can be many microns, with collisions nucleating pinned nanocaps which have a specific shape given by surface energies. The rate-limiting step is found to be the light-activated plucking of the atoms from the substrate. By seeding the nanocaps with spherical Au nanoparticles, growth rates can be compared between different molecules. This reveals a strong influence of bidentate N-coordination of the laterally diffusing metal atoms, which can be frozen out by pretreatment with nickel (II) ions.

Many molecules bound to Au or Pd completely resculpt the surface under even ambient light. This has very significant implications for the preparation of samples, for instance in molecular electronics, catalysis, or sensors. It may account for many unusual phenomena previously observed, but also offers opportunities in optical-based lithography and nano-patterning. A key aspect to explore is the potential for independent control of the lateral migration and the seeding process.

## Methods

### Fabrication

Template stripped Au is used for the substrates, through deposition of 100 nm Au on Si wafers (Si-Mat) by thermal evaporation (NanoPVD-T15A, Moorfield Nanotechnology) and transfer of the Au onto glass (10 mm × 5 mm × 0.55 mm, UQG Optics) using a UV glue (Norland 81)[12,41]. Evaporated Au substrates are made by depositing 3 nm Cr and then 100 nm Au on to Si wafers. The substrates are then immersed for 16 h into 1 mM molecular solutions of each molecule: 2,2'-BPD/BPD (Cambridge Display Technology (CDT), 99.7%), 3,3'-BPD(CDT, 84.85%), BPT(Sigma-Aldrich, 97%), BPDT, Sigma-Aldrich, 95%), 1,4-benzenedimethanethiol (BDMT, Sigma-Aldrich, 98%), MBP(Sigma-Aldrich, 97%), benzene-1,4-dithiol (BDT, Sigma-Aldrich, 99%) in anhydrous ethanol (Sigma-Aldrich, >99.5%), to form the SAM. These are washed in ethanol and blown dry under nitrogen. For samples with nanoparticles, 80 nm diameter synthesized Au nanoparticles were mixed with 0.1 M $NaNO_3$ (10:1), drop cast onto the samples, and washed with DI water for 10 s to form plasmonic NPoM cavities. All fabrication steps are performed in the absence of ambient light. The monolayer of palladium was grown using literature methods[42]. Samples with extra $Ni^{2+}$ were produced by immersing samples in a 50 mM aqueous solution of $NiCl_2$ for 3 h[18].

### SAM characterization

Ensuring the quality and orientation of these SAMs is vital given their pivotal role. The solvent chosen during the preparation of SAMs can profoundly influence molecular orientation, potentially leading to distinct phases at the metal interface, including 'standing up' and 'lying down' configurations[43,44]. The methodology used here guarantees the formation of standing-up BPD SAMs. This is confirmed by both X-ray photoelectron spectroscopy (XPS) spectra (Supplementary Fig. 14) and comparisons with the BPT SAM, whose configuration has been extensively characterized in prior research[33,45].

### DF and Raman spectroscopy/microscopy

DF illumination (Olympus BX51) is used with an incoherent white light source (Philips 7023 12 V 100 W) to locate and characterize NPoM cavities through their scattered light. DF scattering spectra of each NPoM cavity are recorded using a fibre-coupled QE Pro spectrometer (Ocean Optics) with a 0.9 NA 100x objective. Raman spectra are recorded with a fully automated custom-built setup, consisting of a dark-field microscope with motorized stages, as well as spectrometers. A 633 nm CW laser is focused onto each NPoM through the same 0.9 NA 100x objective. The laser power was kept at 5 μW unless stated otherwise.

### DFT calculations

DFT calculations are carried out with the software package Gaussian 16 using the hybrid exchange-correlation functional B3LYP. For C, H, N and S atoms, we use the basis set 6-31 + G(d,p), and for Au atoms, the basis set LanL2DZ. Grimme's empirical D3 dispersion correction was also employed. After geometry optimization, we calculate the Raman activity of BPD which has either one side or both thiol groups bound to a single Au atom (Fig. 4f). Time-dependent DFT calculations were performed with the ground-state optimized geometry and 60 excited states, with a 50:50 singlet to triplet excitation mix.

### XPS

XPS analysis was conducted at the Diamond Light Source synchrotron facility (Oxfordshire, UK), on the I09 beamline. Soft X-ray photons with energy of 700 eV were utilized to examine both unexposed and exposed 2,2'-BPD samples. The samples comprised a BPD SAM on a 100 nm gold layer, supported by a 3 nm chromium adhesion layer on a silicon substrate. The unexposed sample was consistently shielded from light, whereas the exposed sample underwent a 10 h preillumination at 9.5 W cm$^{-2}$ before X-ray analysis. Spectral data were processed using CasaXPS (www.casaxps.com), with binding energy calibration based on the Au 4f7/2 peak at 83.96 eV.

## Data availability

The data that support the findings of this study are available via the Cambridge Open Data archive[46] and from the corresponding author upon request.

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

## Acknowledgements

The authors acknowledge financial support from Cambridge Display Technology Ltd., the European Research Council (ERC) under the Horizon 2020 Research and Innovation Programme THOR (829067), POSEIDON (861950) and PICOFORCE (883703), and UK EPSRC EP/X037770/1 and EP/L027151/1. We thank Diamond Light Source for access to beamline I09 (proposal number SI34784-1) which contributed to the results presented here.

## Author contributions

C.G., P.B., B.d.N. and J.J.B. led the experimental design and sample fabrication. R.A., R.R.R. and M.P.R. provided XPS support, while H.B. synthesized the molecules. SEM measurements came from E.M. and G.D., with S.H. preparing Au and Pd nanoparticles. S.H. and E.E. contributed to experimental code. Simulations and models were developed by P.B., R.A., C.G. and J.J.B. Data analysis was primarily managed by C.G., P.B., B.d.N. and J.J.B., with critical insights from all coauthors. The manuscript was collaboratively written and edited by all authors.

## Competing interests

The authors declare no competing interests.
