## [Peer Review File · Nature Communications]

Extensive photochemical restructuring of molecule-metal surfaces under room lightREVIEWER COMMENTS

Reviewer #1 (Remarks to the Author):

In their present manuscript, J Baumberg and colleagues investigate the photostability of the self-assembled monolayer (SAM) interface under ambient light conditions, focusing on specific molecules and metals. Their study employs optical spectroscopy to track the photoinduced migration of gold atoms within various types of dithiol molecules that are self-assembled on a gold substrate. The authors utilize a combination of experimental characterization techniques, including optical spectroscopy and density functional theory (DFT), to assess the SAM's photostability. Furthermore, they employ optical spectroscopy to monitor the nucleation and growth of gold nanoparticles. Based on their findings, the authors propose a mechanism for the observed photo-migration phenomenon, suggesting that it initiates with the detachment of adatoms from the gold substrate interface molecules, followed by their migration towards the gold-air interface. Additionally, the authors identify specific chemical groups that facilitate this phenomenon.

There are significant concerns regarding the substrate quality and the method used for preparing the self-assembled monolayer (SAM) discussed in this paper. Given the detailed explanations provided below, I cannot recommend the publication of the present article in its current form, unless significant revisions are made.

Comment (1):

The quality of the substrate is of great importance in this study. The authors initiated their research by depositing a 100nm thick layer of gold (Au) onto Si wafers (Si-Mat) through thermal evaporation. Subsequently, they transferred the Au layer onto glass using a UV glue (Norland 81). However, concerns arise regarding the quality of the Au thin film on the polymeric holder. There is a suspicion that the interaction between the gold and the organic polymer might be weak, leading to potential changes at the interface under ambient light conditions. This instability at the interface between the gold and the polymer raises doubts about the claim made by the authors regarding the migration of Au atoms from the gold interface to the molecules in the air interface. It is suggested that the observed nanocaps in the SEM images (Fig 1(d,e)) are more likely due to the deformation occurring at the Norland 81-gold interface rather than the migration of Au atoms.

Comment (2):

To address the potential influence of the gold interface quality in this study, I propose the utilization of alternative substrate options. These options include using Au disposed on a Si-Ti interface, Gold on Mica, or single crystal gold (111). By employing different substrates, it would be possible to minimize the impact of the gold interface quality and focus on investigating the specific phenomena of interest. This approach would provide valuable insights into the factors influencing the observed results, allowing for a more comprehensive analysis.

Comment (3):

The quality of the self-assembled monolayers (SAMs) is a critical aspect of this study. The authors immersed the substrates in 1mM molecular solutions of dithiols in 200 proof anhydrous ethanol for a duration of 16 hours to facilitate the formation of the SAMs. It is widely recognized that the choice of solvent in the preparation of SAMs can significantly impact the behavior of dithiol molecules. Previous research has demonstrated that SAMs prepared using ethanol exhibit two distinct molecular phases at the interface, namely the standing up phase and the lying down phase (1- On the self assembly of short chain alkanedithiols. <https://doi.org/10.1039/B809760G>. 2- Self-assembly of 1, 4-benzenedimethanethiol self-assembled monolayers on gold <https://doi.org/10.1021/la904317b>. 3- Self-assembly of alkanedithiols on Au (111) from solution: effect of chain length and self-assembly conditions <https://doi.org/10.1021/la901601z>). Therefore, it is of utmost importance to thoroughly

analyze the quality of the SAMs, with a particular focus on utilizing XPS spectroscopy to monitor the characteristics of the S2p spectra. This spectroscopic analysis will provide valuable insights into the behavior and composition of the SAMs, allowing for a comprehensive assessment of their quality.

Reviewer #2 (Remarks to the Author):

The manuscript by Baumberg et al. entitled "Extreme photochemical reorganisation of molecule-metal surfaces under room light" presents a novel surface reconstruction process on Au. Metal surface reconstruction under ambient light condition is highly relevant to heterogeneous catalysis and molecular devices. The authors propose a photo-migration mechanism to explain the observed phenomenon. While this paper is certainly innovative, the experimental design could benefit from further improvement to ensure adequacy. After revision of the manuscript the work is suitable for Nature Communications.

In the photoactivity model proposed by the authors, Au adatoms are extracted and then move laterally at the surface under light induction. However, the direct evidence of the formation of Au nanocaps above SAM is missing. It's written in the manuscript (page 3) that the nanocaps have poor conductivity to the metal substrate but the data are not shown. Figure 1f shows the AFM profiles of the nanocaps. They are between 10-20 nm in height (z-dimension). Thus, their plasmonic coupling with the underlying should be strong enough to produce high SERS from the molecular SAM in the gap. The authors can use SERS to confirm the SAM is in the gap between nanocap and substrate film. The DF scattering spectra of NPoMs exhibit a redshift and an increase in intensity, which has been attributed to an increase in the nanoparticle facets. Therefore, it is necessary of the surface nanoparticles' imaging characterization before and after light irradiation to observe the change in nanoparticles morphology, which is a direct evidence of the formation of nanocaps.

In the control experiments, Ni²⁺ cations are used to occupy the coordination sites of N, preventing the generation of nanocaps after illumination. This does not provide evidence that Au and N form coordination bonds. Furthermore, what about the coordination sites for Au atoms in the SAM of MBP or BPDT which do not have bipyridine site? Again SERS of the SAM may provide the evidence of Au-SAM interaction at molecular level.

Reviewer #3 (Remarks to the Author):

The authors present an investigation of the light-induced restructuring of gold-SAM interfaces, showing that relatively low-energy light is sufficient to drive reconstruction of these materials to form gold nanoislands. The work is nicely supported by a range of experimental techniques. I view the results to be highly novel and of interest to a wide audience, although it is somewhat unclear how broadly relevant the observations may be (as opposed to being specific to this particular metal-adsorbate combination). I offer some short suggestions for improving the manuscript:

To what extent are these results limited to the specific molecules shown here? The extent to which this phenomenon can be generalized, if at all, is unclear, given the role these molecules play in the Au diffusion mechanism as well as the highly varied nature of adsorbate-surface interactions.

The last statement of the abstract is too broad: reconstruction may be advantageous in specific examples of catalysis, as gold that is nearly atomically-dispersed may be active when bulk-like gold is not. However, it is incorrect to generalize that reconstructions are helpful for catalysis, as the nature of the ideal active site varies immensely based on the material, application, etc.

I am curious about the saturation conditions and limiting behavior of these studies, particularly given the symmetry of the BPD. If allowed to continue to grow, what is the limiting behavior of the nanocaps?

Very minor points:

The Weibull PDF itself should not contain a differential "dx" term.

The authors may wish to correct a formatting error in the supplementary information table of contents.

In the third line below Figure 1 (Page 3), the authors comment on conductivity. It is unclear how conductivity is being inferred at the metal-molecule interface from SEM/AFM images; is this a typo?

Reviewer #4 (Remarks to the Author):

This interesting study investigates the photoinduced reconstruction of gold-molecule interfaces and the growth of gold nano-islands (in the manuscript called nanocaps) on SAMs on atomically flat gold flakes for seven different molecules. These are BPD, BPT, MBP, BDMT, BDT, BPDT in NPoMs, as well as for BPD with Pd-coated Au, and with Ni²⁺ pre-treatment. Common to all these molecules is that they are bound to the surface via a sulfur atom, forming a well-known chemical bond to gold. The molecules have similar structures and are either based on a 2,2'-bipyridin, a 3,3'-bipyridin core, a biphenyl-, or a phenyl core, the linker to gold is either a mercaptomethyl- or a mercapto-group. The growth kinetics of the nanocaps for the different surfactants is systematically investigated under white light illumination, by a confocal optical microscope with an attached Raman-spectrometer for up to 60 hours. Repeated image and spectra collection over such an impressively long time period is made possible by an automated sample translation and custom software. The gold nano-islands are characterized by white light scattering spectroscopy SEM- and AFM-imaging, and the molecules are characterized by SERS-spectroscopy. AFM profiles reveal that nanocaps can rise by more than 20 nanometres above surface. The spectral behavior of the nanocaps is further contrasted to nanoparticle-on-mirror (NPoM) plasmonic cavities by placing Au nanoparticles (diameter 80nm) on top of the SAM-coated Au showing red shifted coupled plasmon modes above 800 nm. The polarizabilities of the molecules is calculated by time dependent density functional theory. The formation and growth of the nanocaps is explained and visualized photoinduced migration of Au-atoms through the SAM. The whole study is impressive and contains a huge amount of information related to large and important field of metal-molecule interfaces.

However, I have a couple of questions concerning the following:

How do the authors exclude that no thermal effects lead to a Au atom migration and nanocap growth?

Concerning the different spectral behavior of the nanocaps vs. the particle-on-mirror nanocavities.

Comparing the DF-spectra of nanocaps (Fig.1c) and particle-on-mirror nanocavities (Figs. 4a and 4b). Why is there only one broad band with a maximum at around 530 nm v.s. like for Au nanospheres, while the nanocaps seem to be elongated? I would also expect to see also a longitudinal plasmon mode.

For particle-on-mirror nanocavities long wavelength bands from coupled plasmon modes are visible, how do the authors explain that similar bands are missing for nanocaps?

Charge transfer from the Au surface to the molecule should affect the aromatic ring vibrations of the molecules. Can a difference be observed in the relative intensities of the respective bands in the spectra for the various compounds?

Are the SERS spectra for one and the same compound different for nanocaps and particle-on-mirror nanocavities in the respective spectral region, e.g. the vibrational modes of the aromatic rings?

Alfred J. Meixner

Response to Reviewers comments:

We are delighted that the reviewers emphasise this advance is ‘*certainly innovative*’, ‘*the work is suitable for Nature Communications*’ (reviewer #2), ‘*highly novel and of interest to a wide audience*’ and ‘*nicely supported by a range of experimental techniques*’ (reviewer #3), and the ‘*whole study is impressive*’ and ‘*contains a huge amount of information related to the large and important field of metal-molecule interfaces*’ (reviewer #4). We now provide extensive new data with control experiments as requested, including XPS, STM, darkfield, temperature dependences, and simulations. We answer all the points below in detail.

Reviewer #1:

1. *The quality of the substrate is of great importance in this study. The authors initiated their research by depositing a 100nm thick layer of gold (Au) onto Si wafers (Si-Mat) through thermal evaporation. Subsequently, they transferred the Au layer onto glass using a UV glue (Norland 81). However, concerns arise regarding the quality of the Au thin film on the polymeric holder. There is a suspicion that the interaction between the gold and the organic polymer might be weak, leading to potential changes at the interface under ambient light conditions. This instability at the interface between the gold and the polymer raises doubts about the claim made by the authors regarding the migration of Au atoms from the gold interface to the molecules in the air interface. It is suggested that the observed nanocaps in the SEM images (Fig 1(d,e)) are more likely due to the deformation occurring at the Norland 81-gold interface rather than the migration of Au atoms.*

> We certainly appreciate this suggestion concerning the influence of UV glue, which is indeed an important consideration. However based on our extensive experimental data, if weak gold-glue binding were a significant factor, we would anticipate surface deformations across all samples. Contrarily, from the >100 samples measured, pronounced gold transport was evident exclusively in only the samples with a few specific molecules, such as 2,2'-BPD. In stark contrast, samples with only BPT or BPDT showed neither gold transport nor any signs of surface deformation under the Au substrate (see Fig.R1 below taken at 0.1 W/cm² white intensity). Alongside the single NP at the image centre (used for tracking), nanocap formation is observed only for 2,2'-BPD. Additional detailed control experiments are shown below, in response to comment 2.

Fig. R1: Evolution over time of samples with monolayers of (a) 2,2'-BPD, (b) BPT, and (c) BPDT illuminated under 0.1 W/cm² white light.

2. To address the potential influence of the gold interface quality in this study, I propose the utilization of alternative substrate options. These options include using Au disposed on a Si-Ti interface, Gold on Mica, or single crystal gold (111). By employing different substrates, it would be possible to minimize the impact of the gold interface quality and focus on investigating the specific phenomena of interest. This approach would provide valuable insights into the factors influencing the observed results, allowing for a more comprehensive analysis.

> As the reviewer suggests, control experiments are useful to show if the gold interface quality is important, and the effect of alternative substrates. We thus show data for an alternative substrate configuration: thermally depositing 100nm Au atop 3nm Cr on a (100) Si undoped wafer (no glue). After similarly coating 2,2'-BPD SAMs on this substrate, the samples are subjected to 10W/cm² white light for 24 hours. As shown below (Fig.R2 and S12), nanocaps clearly form and grow on this sample exactly as before, showing the glue cannot have any influence. We thus add this data to the SI, and comment briefly in the main text.

Fig. R2: Time-evolution of 2,2'-BPD on Au-coated Si substrate under white light exposure. Changes observed in 2,2'-BPD sample under 10W/cm² white light over 24 hours. Panels (a-e) show sample evolution captured with 20x objective, (f-j) with 100x objective.

3. The quality of the self-assembled monolayers (SAMs) is a critical aspect of this study. The authors immersed the substrates in 1mM molecular solutions of dithiols in 200 proof anhydrous ethanol for a duration of 16 hours to facilitate the formation of the SAMs. It is widely recognized that the choice of solvent in the preparation of SAMs can significantly impact the behavior of dithiol molecules. Previous research has demonstrated that SAMs prepared using ethanol exhibit two distinct molecular phases at the interface, namely the standing up phase and the lying down phase (1- On the self assembly of short chain alkanedithiols. <https://doi.org/10.1039/B809760G>. 2- Self-assembly of 1, 4-benzenedimethanethiol SAMs on gold <https://doi.org/10.1021/la904317b>. 3- Self-assembly of alkanedithiols on Au (111) from solution: effect of chain length and self-assembly conditions <https://doi.org/10.1021/la901601z>). Therefore, it is of utmost importance to thoroughly analyze the quality of the SAMs, with a particular focus on utilizing XPS spectroscopy to monitor the characteristics of the S2p spectra. This spectroscopic analysis will provide valuable insights into the behavior and composition of the SAMs, allowing for a comprehensive assessment of their quality.

> We fully agree with the reviewer that SAM quality and orientation are important. Indeed our prior work confirms we produce the standing-up phase (as is well known for these longer immersion times, since it is favoured at higher thiol binding density [see 10.1021/la901601z]). This is why we use 16-hour immersion and high molecular concentration, as now clarified in the Methods.

XPS measurements are now performed as suggested (Fig.R3, and new Fig.S14), revealing a doubling in the peak ratio of Au-S:free-SH post-illumination (from 0.7 to 1.6), indicative of additional SH binding to Au, which is consistent with nanocap formation and also not suggestive of a lying down molecular phase.

Fig. R3. Sulphur 2p XPS spectra of BPD SAMs before and after light exposure. (a) BPD SAMs after 16h immersion in 1mM ethanolic solution, (b) BPD SAMs after exposure to 10W/cm² white light for 10h.

While XPS can give some information, here we also use optics and STM to directly probe the SAM, using both the darkfield scattering resonances, as well as the Raman spectra. The scattering resonances at $t=0$ (Fig.4a, S5) depend on the NP diameter, gap size and gap contents (see [32], and also 10.1021/acsphotonics.2c00116), which give gap sizes ~ 1.3 nm for BPT. This is confirmed by taking new ambient STM measurements on this SAM (Fig.R4 below), showing the known lattice for the standing-up molecular phase, with indeed the expected thickness of 1.3nm [see 10.1021/nl5041786].

Fig. R4. STM analysis of BPT SAM on template-stripped Au. (a) STM image, size 15nm, after background removal and Fourier filtering, $I_t = 0.3$ nA and $V_t = +0.3$ V. (b) 2D-FFT of STM image in (a). Red circles show regions of the 2D-FFT used to generate the Fourier filtered image shown in (c), with unit cell marked.

Further information is provided by comparing darkfield spectra of BPT (mono-thiol) and 2,2'-BPD (di-thiol). Our data show a slightly blue-shifted coupled mode position for BPD relative to BPT (Fig.R5 below). This implies a slightly larger gap between the AuNP and Au substrate for BPD compared to BPT, which matches expectations given the slightly larger molecular length of BPD. Using our full electromagnetic simulations, we estimate the BPD height as ~ 1.65 nm, as also expected for standing-up SAMs with near-normal orientation. Heights for both BPT and BPD thus correspond well with the bond distances between their outermost atoms. We thus incorporate this discussion into the main text and SI (Fig.S15).

Fig. R5: Gap size analysis of BPT and 2,2'-BPD SAMs. Experimental (solid line) and simulated (dashed) scattering spectra comparing BPT (black) and 2,2'-BPD (red) self-assembled monolayers in NPoMs. For simulations, gap sizes of 1.3 nm for BPT and 1.65 nm for BPD are used.

Reviewer #2:

1. *In the photoactivity model proposed by the authors, Au adatoms are extracted and then move laterally at the surface under light induction. However, the direct evidence of the formation of Au nanocaps above SAM is missing. It's written in the manuscript (page 3) that the nanocaps have poor conductivity to the metal substrate but the data are not shown.*

> The reviewer highlights this desirable information. Using top-view SEMs (Fig.1d), we observe dark nanocaps that are irregular in shape and range from 50-400 nm in width. Voltage contrast [see 10.1016/0304-3991(94)00183-N, 10.1116/1.589811, 10.1039/C1NR10512D, 10.1063/1.2207552, 10.1088/1361-6501/aaeab8, 10.1016/j.ultramic.2005.03.007, 10.1088/0957-4484/26/8/085703, now added] implies these darker nanocap regions are areas of lower secondary electron emissivity, related to reduced conductivity. This implies that the nanocaps are less electrically connected to the Au substrate, hence our inference that a continuous molecular layer remains underneath.

2. *Figure 1f shows the AFM profiles of the nanocaps. They are between 10-20 nm in height (z-dimension). Thus, their plasmonic coupling with the underlying should be strong enough to produce high SERS from the molecular SAM in the gap. The authors can use SERS to confirm the SAM is in the gap between nanocap and substrate film.*

> Indeed this is an excellent idea. We tried this extensively, but no significant SERS is observed. The problem is that the observed resonances at ~540nm (Fig.1c) are far detuned from the laser (providing very weak confinement), while the flat geometry of the nanocap (Fig.1e) is extremely poor at coupling light into the gap mode (with vertical field orientation, perpendicular to the incident light). Simulations to confirm this are shown in the response to reviewer #4 point 2. The very low in/out coupling minimises SERS signals, and higher laser powers introduce additional issues. For instance irradiation of the nanocaps appears to 'evaporate' them again (Fig.R5), with powers exceeding 10 μ W for 447nm excitation leading to the systematic reduction in DF scattering amplitude and reversal of the red shift seen in Fig.1c.

3. *The DF scattering spectra of NPoMs exhibit a redshift and an increase in intensity, which has been attributed to an increase in the nanoparticle facets. Therefore, it is necessary of the surface nanoparticles' imaging characterization before and after light irradiation to observe the change in nanoparticles morphology, which is a direct evidence of the formation of nanocaps.*

> This is also a nice idea, but measuring facet sizes is challenging. The e-beam in SEM induces extra cross-linking of BPD molecules (see [18]), making direct before-and-after imaging problematic. We thus adopt an alternative, comparing BPD SAM-exposed samples with those without any SAM between the Au NP and Au substrate. Focused Ion Beam Scanning Electron Microscopy (FIB-SEM)

images (Fig. R6 below, S4 in paper) suggest that the Au NP facet size post-exposure on the BPD SAM is notably larger than in the control, supporting this Au migration hypothesis.

Fig. R6: Focused ion beam scanning electron microscopy (FIB-SEM). After the NPoM is overcoated with Pt using electron beam deposition, etching by the FIB gives cross sectional views. (a) Au NP directly on Au mirror, and (b) Au NP on Au mirror spaced with 2,2'-BPD SAM after exposure to light.

4. *In the control experiments, Ni²⁺ cations are used to occupy the coordination sites of N, preventing the generation of nanocaps after illumination. This does not provide evidence that Au and N form coordination bonds. Furthermore, what about the coordination sites for Au atoms in the SAM of MBP or BPDT which do not have bipyridine site? Again SERS of the SAM may provide the evidence of Au-SAM interaction at molecular level.*

> This discussion of the control experiments is very helpful. Our aim for the Ni²⁺ control is to show that occupation of the N-coordination sites turns off the Au migration. Indeed we agree this does not prove Au-N coordination, but shows the strong requirement that this N coordination is present. We note that nanocap formation on flat Au occurs only for 2,2'-BPD where the coordination is present, but separately show that a restructuring of Au under NPoMs is a more general phenomenon for a wider range of molecules. Under more extreme conditions MBP starts to show nanocap-type effects, so clearly a more general coordination may be possible. We thus clarify these differences in the main text.

Reviewer #3:

1. *To what extent are these results limited to the specific molecules shown here? The extent to which this phenomenon can be generalized, if at all, is unclear, given the role these molecules play in the Au diffusion mechanism as well as the highly varied nature of adsorbate-surface interactions.*

> While we studied an initial broad set of molecules, so far the key feature identified for nanocap formation on flat Au is the N-coordination. We emphasise this is different from our comparison geometry which uses the NP-on-mirror to focus on the late stage of the process when seeded with an 'artificial' nanocap of spherical shape. In the latter case, different molecules indeed produce effects from a wider variety of adsorbate-surface interactions as the reviewer notes. In the former case on flat Au which is the key novel aspect, we compare: (i) type of ring system (aromatic vs bipyridine), (ii) position of N within bipyridine rings (2,2' vs 3,3'), (iii) number of aromatic rings (BDT vs BPDT), (iv) methylene groups (BPDT vs MBP, BDT vs BDMT), and (v) mono- vs di-thiols (BPT vs BPDT). Overall the 2,2'-BPD has vastly stronger activity, which suggests the key interaction is Au-N. Our aim is that this insight stimulates further experimental tests of this strong photochemistry.

2. *The last statement of the abstract is too broad: reconstruction may be advantageous in specific examples of catalysis, as gold that is nearly atomically-dispersed may be active when bulk-like gold is not. However, it is incorrect to generalize that reconstructions are helpful for catalysis, as the nature of the ideal active site varies immensely based on the material, application, etc.*

> We fully agree this overgeneralization is incorrect. We thus revise the abstract to present a more nuanced perspective on the advantages and disadvantages of surface reconstruction.

3. *I am curious about the saturation conditions and limiting behavior of these studies, particularly given the symmetry of the BPD. If allowed to continue to grow, what is the limiting behavior of the nanocaps?*

> Indeed this is an intriguing question about the limiting behavior of the nanocaps. On flat Au, our most extended observation period spans 5 days and as time progresses we observe some reduction in the rate of new seeding of nanocaps (N_0) but little change in the growth rate of each one once formed (κ , Fig.2c). Eventually interactions between nanocaps must become more important. By contrast, saturation in the NP-on-mirror red-shift over time is always observed (Fig.S5,6), after the facet expands to the maximum allowed by the NP diameter.

4) *Very minor points:*

The Weibull PDF itself should not contain a differential “dx” term.

> The reviewer helpfully spots an evident error, now corrected in the manuscript.

5) *The authors may wish to correct a formatting error in the supplementary information table of contents.*

> We appreciate this observation and changed the Supplementary information appropriately.

6) *In the third line below Figure 1 (Page 3), the authors comment on conductivity. It is unclear how conductivity is being inferred at the metal-molecule interface from SEM/AFM images; is this a typo?*

> As also noted by reviewer #2 (point 1), we did not explain this clearly enough. As discussed above, darker nanocap regions in SEM are areas of lower secondary electron emissivity, related to reduced conductivity, and implies that the nanocaps are not electrically connected to the Au substrate.

Reviewer #4:

1. *How do the authors exclude that no thermal effects lead to a Au atom migration and nanocap growth?*

> Indeed it is important to understand potential thermal effects. Our previous work [ref 12] found that surface temperatures of these samples for 10W/cm² halogen light exposure (10⁻⁴ μW/μm²) remain at ~300K (see Fig.R7a below). As a control, when samples are stored at 28°C in darkness for a week, we observe no alterations on the surface. To delve deeper into the influence of elevated temperatures, we conduct two distinct experiments with identically prepared samples. One sample is exposed to temperatures of 100°C in a dark environment for 10 hours. While a slight increase in surface roughness is seen, nanocaps do not form (Fig.R7b). In contrast, when the other sample is exposed to 10W/cm² light, nanocaps appear within 4 hours (Fig.R7c). This strongly implies that while temperature can influence surface morphology, the observed nanocap growth is not due to thermal effects. We add a discussion of this in the main text, and add this figure to the SI as Fig.S13.

Fig.R7: Thermal effect on BPD molecules on Au. (a) Temperature measured by anti-Stokes SERS vs laser power (from [12]). Reproduced from Ref [12] by Lin, Q., et al. Optical suppression of energy barriers in single molecule e-metal binding, *Sci. Adv.* 8, *Sci. Adv.* 8, eabp9285 (2022), CC BY. (b) BPD sample image before and after 10h at 100°C. (c) BPD sample before and after 4.5h of 10W/cm² white light exposure.

2) Concerning the different spectral behavior of the nanocaps vs. the particle-on-mirror nanocavities. Comparing the DF-spectra of nanocaps (Fig.1c) and particle-on-mirror nanocavities (Figs. 4a and 4b). Why is there only one broad band with a maximum at around 530 nm v.s. like for Au nanospheres, while the nanocaps seem to be elongated? I would also expect to see also a longitudinal plasmon mode.

> Indeed as the reviewer notes, there is a very significant difference in spectral behavior between nanocaps and particle-on-mirror nanocavities. The key distinction lies in the morphology of the nanocaps. As well their elongated shape they are quite flat, which prevents access to nanogap coupled modes as obtained from taller nanoparticles. We validate this using finite-difference time-domain (FDTD) simulations, comparing an 80nm nanoparticle (NP) and a nanocap with dimensions of 75nm in width and 20nm in height. The scattering spectra from experiment and these simulations are compared in Fig.R8 below, and confirm that indeed the longitudinal plasmon mode is seen only for the NPs ($E \perp$ metal). We now add this figure to the SI as Fig.S16 and discuss it.

Fig. R8: Scattering spectra comparing nanocaps and NP particles. (a) Experimental darkfield spectra, and (b) FDTD simulations comparing NPoM of 80nm Au (red) with nanocap of 75nm diameter and 20nm height (black), gaps of 1.5nm.

3) For particle-on-mirror nanocavities long wavelength bands from coupled plasmon modes are visible, how do the authors explain that similar bands are missing for nanocaps?

> As noted in (2) above, the flattened morphology of nanocaps (which are much thinner than the resonant optical wavelength) dramatically reduces the in/out-coupling, meaning that although gap modes can exist, they cannot be observed. The key reason is that for flat structures, the optical field is only in-plane, and thus cannot excite the out-of-plane mode in the nanogap (Fig.R8b).

4) *Charge transfer from the Au surface to the molecule should affect the aromatic ring vibrations of the molecules. Can a difference be observed in the relative intensities of the respective bands in the spectra for the various compounds?*

> Indeed, the SERS spectra for each molecular SAM in the NPoM show shifts in spectral positions and relative intensities compared to Raman (solution or powder), as has been documented in many previous works. These effects are due to different tilt angles and intermolecular interactions of molecules in the SAM as well as charge transfer with the Au (as the reviewer notes). It is however not yet possible to specifically associate different shifts with charge transfer, especially because image charge effects in the nearby metal also shift the SERS modes, and solvation effects are modified in the constrained environment.

Comparing SERS of BPT and BPD SAMs reveals distinct peak intensity differences. BPD shows on average tenfold greater SERS than BPT (Fig. R9a) while the gap size difference ($\text{SERS} \propto d^{-2}$) would account for a factor 1.7 [ref 32]. Extra effects come from the difference in Raman cross section of the aryl vs pyridine rings as well as possible charge transfer (see [12]). There are also changes in the peak ratios (Fig. R9b), which can arise from molecular orientation.

A significant challenge is the inadequacy of DFT in precisely accounting for SERS spectra in such nanogaps. Even for simple non-redox molecules (such as BPT), state-of-the-art DFT does not recover the observations (due to the points above). Thus it is not yet in a position to prove if charge-transfer is involved in the BPD mechanism. We now briefly discuss these aspects in the manuscript.

Fig. R9: SERS spectra comparing BPT and BPD NPoMs. (a) Raw experimental data (averaged over 50 NPoMs) and (b) after normalization for easy comparison.

5) *Are the SERS spectra for one and the same compound different for nanocaps and particle-on-mirror nanocavities in the respective spectral region, e.g. the vibrational modes of the aromatic rings?*

> Indeed this would be very useful to discern, but as noted for reviewer #2 point 2, we do not obtain SERS from the nanocaps. The reason is as shown in the simulations of Fig.R8 and point (2) above, because nanocaps do not couple far-field light into the nanogaps. As the aromatic rings are orthogonal to the longitudinal plasmon field, any SERS is too weak to observe.

REVIEWERS' COMMENTS

Reviewer #1 (Remarks to the Author):

The authors have addressed every issue I raised. The paper now contributes novel insights in the field of molecular self-assembly monolayers and nanofabrication. They employ a robust combination of techniques for characterizing this phenomenon. From analysis to interpretation, the methodology adheres to the field's benchmarks for Self-Assembled Monolayers.

Reviewer #2 (Remarks to the Author):

The authors revised the manuscript according to reviewers' comments, with detailed explanation and reasonable experimental design. Generally, the manuscript has been improved. I have one further question about the lacking of direct evidence of N coordination. The authors said in the response that the experiment indeed does not prove Au-N coordination, but shows the strong requirement that this N coordination is present. In the revised manuscript, the Au-N coordination is still the mechanism behind Au restructuring. For example, in paragraph 8, "We suggest that for 2,2'-BPD, the diffusive sites are N-coordinated by the molecules, as previously seen for Ni²⁺". Since Au-N coordination is not experimentally supported, it needs to be further clarified in the main text of manuscript. After this minor modification, this work is suitable for publication in Nature Communications.

Reviewer #3 (Remarks to the Author):

The authors have satisfactorily addressed my comments, and I believe the manuscript to be suitable for publication.

Reviewer #4 (Remarks to the Author):

The authors have carefully addressed my concerns by doing additional experiments or providing additional data. They have revised their manuscript accordingly. Hence, I suggest publication of the revised manuscript and supplementary.